# Enhanced Sensing and Data Processing System for Continuous Profiling of Pavement Deflection

**DOI:** 10.3390/ma12101653

**Published:** 2019-05-21

**Authors:** Boo Hyun Nam, Kenneth H. Stokoe, Heejung Youn

**Affiliations:** 1Department of Civil, Environmental, and Construction Engineering, University of Central Florida, Orlando, FL 32816, USA; 2Department of Civil, Architectural, and Environmental Engineering, The University of Texas at Austin, 301 E. Dean Keeton St. Stop C1700, Austin, TX 78712, USA; k.stokoe@mail.utexas.edu; 3School of Urban and Civil Engineering, Hongik University, 94 Wausanro, Mapo-gu, Seoul 04066, Korea

**Keywords:** continuous deflection profile, rolling dynamic deflectometer, rolling sensor, data processing

## Abstract

Continuous pavement deflection profiling in a nondestructive manner has received great attention because of its efficiency in pavement evaluation. The Rolling Dynamic Deflectometer (RDD) is a continuous pavement deflection profiling technology and has demonstrated its successful use at many pavement projects. However, RDD’s current test speed of 1.6 km/h (with the prototype rolling sensor) often limits its use in large-coverage projects and traffic congested areas. Increasing the test speed creates a higher-noise environment, lowered signal-to-noise ratio, and sensor decoupling with the ground surface. This study presents the enhancement to the RDD for increased test speed, associated with the new design of the lower-level rolling sensor and higher-performance digital filter and data processing. The new sensor along with the enhanced data processing could increase the spatial resolution of the deflection data, which allows the increase of the test speed of the RDD.

## 1. Introduction

Deflection-based nondestructive testing (NDT) methods are popular for pavement evaluation. Particularly, the Falling Weight Deflectometer (FWD) is the most common testing method. The FWD is operated at a stationary mode that causes a limited coverage of a tested path for a given time. In addition, traffic interruption due to FWD’s stop-and-go operations can cause traffic hazards and an unsafe work environment as well. In order to continuously measure pavement deflections at higher speeds, extensive efforts have been carried out over the past decades. The continuous deflection measurement technologies currently in use are the Traffic Speed Deflectograph (TSD) [1,2,3,4,5], Rolling Wheel Deflectometer (RWD) [6,7,8,9,10], Quest Integrated/Dynatest Consulting Rolling Weight Deflectometer [11,12], Road Deflection Tester (RDT) [13,14], and the Rolling Dynamic Deflectometer (RDD) [15]. In January 2018, Dynatest Raptor (Rapid Pavement Tester) was introduced to the Transportation Research Board (TRB). The RAPTOR uses the Dynatest RWD (Rolling Weight Deflectometer). The requirement for the length of the measurement beam is significantly reduced by expanding the measurement system with highly accurate inertial sensors for tracking beam movements [16]. Details of the five testing devices such as the testing operational principle, loading and sensing mechanisms, and an example of the field data can be seen in each of the references above. Particularly, Rada and Nazarian (2011) published a report entitled “The State-of-the-Technology of Moving Pavement Deflection Testing” [10]. Based on literature, TRB Workshop 144 during the 88th Annual TRB Meeting (title: *High Speed Pavement Deflection Measurements*), and information from the equipment manufacturers, Rada and Nazarian [10] identified the following three devices as viable tools: TSD, RWD, and RDD. The other two devices, which are the Quest/Dynatest RWD and Swedish RDT, are not recommended because continuous research was either reduced or stopped. The prototype equipment may not be available. Table 1 summarizes the key descriptions of each testing method. As seen in the table, the RDD provides the best spatial data resolution but slowest test speed in comparison with other devices. 

It is important to note that the RDD is basically similar to the operation principle of the Falling Weight Deflectometer (FWD), but with the sinusoidal loading. Other continuous contactless devices adopted the test principle of the Benkelman Beam test, which is vertically static loading, but a moving load. Unlike others, the RDD employs contact-type loading (with a sinusoidal loading) and a sensing system so that the accuracy and data spatial resolution are superior to those contactless technologies (with non-contacting sensors). Those contactless equipment employ a moving loading system (not sinusoidal loading), thus no use of a digital band-pass filter that can remove unwanted noise signals. The data spatial resolution ranges from 33 to 100 ft (up to 500 ft), which is not sufficient for project level studies, thus they are used only for network-level studies. On the other hand, the RDD, with higher accuracy and spatial resolution, is mainly used for project-level studies. In other words, although the technologies listed in Table 1 are all continuous testing devices, but the main purposes of the RDD and other continuous contactless equipment are different. The RDD has been extensively used in TxDOT’s project-level studies, for example pavement forensic investigation, identification of poor areas (or slabs) to be repaired, joint load transfer assessment, selection of optimum rehabilitation strategy, and so on [17,18,19,20,21,22,23]. 

The current testing speed of the RDD is about 1.6 km/h (1 mph). The RDD is designed to overcome FWD’s coverage limitation by producing a continuous pavement deflection profile [15]. During the last two decades, many pavement projects have demonstrated benefits of the RDD as both a screening and evaluation tool; for example, (1) delineation of problematic areas to be repaired [15,19,20], (2) pavement forensic investigation [17,18,19,20], (3) monitoring changes of pavement behavior [21], (4) load-transfer evaluation at cracks and joints [22,23,24], (5) accelerated pavement testing to estimate the remaining life [25,26], and (6) selection of optimum rehabilitation treatments [18,19,22,23].

This study presents the enhanced sensing and data processing system for newly developed rolling sensor (speed-improved sensor). The new sensor was designed for the target speed of 8.0 km/h and also for the measure of more “point” deflection that is suitable for load-transfer assessment of transverse cracks and joints. Subsequently, new data processing was developed for the target speed of 8.0 km/h; more specifically, aiming at improving spatial resolution of deflection data and improving the signal-filtering performance. The new data processing scheme has adopted the following three methods: distance-based deflection reporting method, shifting technique of moving average method, and continuous signal-to-noise ratio and noise profiles. 

## 2. Background on the RDD

### 2.1. The RDD

The RDD is a truck-mounted device that consists of the three systems: a system of dynamic loading and force measurement, a sensor system, and a distance recording system. Figure 1 shows a schematic diagram of the RDD and the arrangement of the rolling sensors. The RDD continuously loads (with a 30 hz sinusoidal loading) pavement through loading rollers as the device moves along the road. Simultaneously, rolling sensors of the RDD, that consists of three wheels and a velocity transducer (2 Hz geophone), continuously measure the induced pavement response (dynamic deflection) at multiple locations. With the original rolling sensors, the RDD is used to profile along the pavement at approximately 1.6 km/h (1 mph). During RDD testing, both static and dynamic forces are applied to the pavement through the two loading rollers. The RDD loading system is capable of generating static forces of 13 to 180 kN (3 to 40 kips) and dynamic sinusoidal forces with a peak-to-peak amplitude of 9 to 310 kN (2 to 70 kips) over the frequency range of about 10 to 100 Hz. In typical highway rehabilitation projects, a static hold-down force in the range of 36 to 45 kN (8 to 10 kips) and a peak-to-peak dynamic force in the range of 36 to 45 kN (8 to 10 kips) at an operating frequency of 30 Hz are used. The original rolling sensor (referred as the first-generation sensor) was developed by Bay and Stokoe [15]. This sensor is a freestanding sensor and composed of three 6 inch diameter wheels. A velocity transducer (2-Hz geophone) is located at the geometric center of the three wheels, as seen in Figure 2a. Typically four rolling sensors, Sensors #1–4 in a linear array (see Figure 1b), measure pavement movements (dynamic deflections) under the sinusoidal loading. Sensor #1 is positioned in between the two loading rollers and provides the maximum pavement deflection because it is closer to the loading points than other sensors. A distance measurement sensor (rotary encoder) is attached on a rear wheel of the truck and the distance moved along the pavement is recorded. The output signals from all three sensing systems are synchronized and processed to produce the continuous pavement deflection profile.

***Limitations:*** The speed of this sensor is limited to 1.6 km/h (1 mph) due to two issues: decoupling of the sensor and high rolling noise from rough surface texture. The sensor wheels tend to lose the contact from the ground at increased rolling speeds due to a lack of hold-down force except self-weight. When the induced acceleration exceeds −1 g, the force between the sensor wheel and the ground is zero. Increased rolling speeds, assuming the sensor is in contact with the pavement surface, dramatically increase the rolling noise. It is reported that the double in the rolling velocity almost doubles the rolling noise in magnitude [27]. In addition, the test speed is manually controlled by a truck driver. Thus, the concern of sensor decoupling always exists due to the speed variation.

### 2.2. RDD Signal Processing

The original data processing of the RDD was designed for the 1.6 km/h test speed [27]. During testing (with a sinusoidal loading to pavement), the rolling sensor, housing a velocity transducer, continuously records induced velocity signals at a 256 sampling rate. The raw data are then multiplied by a complex function. The complex demodulation products are then filtered by a digital band-pass filter. The complex time series after filtering are then multiplied by the sensor calibration factor. The calibrated time series that are velocity values are then converted to peak-to-peak displacements (referred as dynamic deflection). The displacement series are then averaged over a selected time interval (typically 1 or 2 seconds). This time-based reporting can be affected by the variation of the rolling (or test) speed. For instance, higher testing speeds cause lower spatial data resolutions. 

The filter design by Bay [27] aimed at separating the RDD signals, which is the induced pavement deflection at a given loading frequency (typically 30 Hz), from other noise signals (referred as rolling noise). The prototype filter design employed a composite infinite-impulse-response (IIR) and finite-impulse-response (FIR) filter, which is a band-pass (notch-pass) filter. The variables that affect the filter performance are (1) filter bandwidth, BW_20_, (2) filter settling time, t_90_, and (3) attenuation in the reject bands. The design of the IIR-type filter involves the positioning of poles and zeros in the Z-domain [28,29]. The transformation occurs from frequency to Z domain by:z = e^−*i*2πf/fs^(1)
where fs is the sampling frequency, f is frequency, and z is a complex-valued variable (in the Z-domain). In general, complex poles and zeros are in complex conjugate pairs. The IIR digital filter response in the Z-domain, H(z), is expressed as below:(2)$$H\left(z\right)=\frac{\left(z-r_{1}\right)\left(z-r_{2}\right)\left(z-r_{3}\right)\cdots \left(z-r_{n}\right)}{\left(z-p_{1}\right)\left(z-p_{2}\right)\left(z-p_{3}\right)\cdots \left(z-p_{n}\right)}$$
where r_n_ is a complex value that corresponds to the nth zero location, H(z) is the filter response, and p_n_ is a complex value that corresponds to the nth pole location.

The width of the filter response at −20 dB is defined as a filter bandwidth, BW_20_. As the pole value approaches 1.0, the filter becomes narrower, and BW_20_ decreases. Decreasing BW_20_ represents the increasing of the filter reject-band attenuation. The filter settling time, t_90_, is another variable affecting the performance of the IIR filter. The time required for the filter to respond to a change in input is defined as a filter settling time. The t_90_ is a parameter that indicates the time for the amplitude of a digital filter to settle 90%. As the filter band width becomes narrower, the noise attenuation increases; however, there is a trade-off between BW_20_ and t_90_. 

The FIR filter with decimation is then added to the IIR filter. The decimation refers to a filtering procedure that fewer points are produced in its output than its input [16]. When the sampling rate of 256 is used, 1 s time interval (typical data resolution) produces 256 data points. There is no benefit to keep all data points over the required time interval, thus averaging the data recorded over the selected time interval would be desirable. The numbers of data points to be averaged is referred as N_avg._, which is the input parameter of the FIR filter. The addition of the FIR filter causes a slightly narrower pass-band and a substantially larger attenuation in the reject bands.

***Limitations***: RDD’s prototype data processing is based on a time-based reporting method that have several limitations summarized below: (1) The speed variation due to a manual speed control by a truck driver can create variations of data in spatial resolution. (2) The increased speed may miss critical features (e.g., cracks, joints) in the deflection profile. A typical t_90_ of 1 or 2 s results in about 0.3 or 0.9 m (1 or 3 ft) distance interval. This issue of missing data points will become more significant as the test speed increases. (3) The current data processing that averages deflection data over a distance interval (controlled by t_90_) underestimates a deflection at joints. At joints, an additional underestimation may occur when the distance interval corresponding to t_90_ does not cover equal portions of two adjacent slabs. (4) In a jointed concrete pavement (JCP), the time-based method has difficulty in locating the positions of rolling sensors at a joint. For proper load-transfer assessment, two sensors should be positioned with equal distance from the joint. For example, Sensor #1 is on the loaded slab and Sensor #2 is across the joint. Inappropriate positions of Sensors #1 and #2 will result in incorrect load-transfer efficiencies (LTEs) of joints. 

## 3. Speed-Improved Rolling Sensor

### 3.1. Sensor Design

In the sensor design, the three design parameters to consider are sensor decoupling, rolling noise, and higher signal-to-noise ratio (SNR). Thus, this study adopted three design criteria for sensor development: (1) hold-down force to prevent the sensor decoupling at higher rolling speeds, and (2) larger-diameter and wider wheels that are less sensitive to surface texture and roughness, thus reducing rolling noise. The mechanical design and hardware fabrication of a speed-improved rolling sensor can be seen in Nam et al. (2014) [30]. It is important to note that relatively compliant wheels may be desirable for the rolling sensor with respect to attenuating rolling noise. A study on the effect of the roller urethane hardness on noise was conducted [30] and the urethane hardness of 70A showed the best performance. As seen in Figure 2b, the new-generation sensor involves two rolling wheels, 30.48 cm (12 in.) diameter and 5.08 cm (2 in.) width, a 2 Hz geophone in the middle, and one large air spring on the top. Unlike the old generation sensors, the two-wheel system measures deflections through two contact points, aimed at capturing critical features such as transverse cracks and joints. 

### 3.2. Evaluation of Sensor Design Parameters

The performance of the rolling sensor was evaluated associated with the three performance criteria: sensor decoupling, the magnitude of rolling noise, and SNR. The sensor performance was tested at varied test speeds up to 8 km/h. The level of negative accelerations on the rolling sensor while rolling is a good indicator to check the sensor decoupling. The theoretical maximum vertical acceleration along with different levels of hold-down force were calculated through Equations (3) to (6) (see below) and summarized in Table 2. For example, the theoretical maximum negative acceleration to maintain the sensor coupling; 34.5 kPa (5 psi) hold-down pressure results in the maximum negative acceleration of 5.2 g.
*F* = *F_hd_* + *W_sensor_* = *ma*(3)
(4)$$W+F_{hd}==\frac{W_{sensor}}{g}a$$
(5)$$a=\left(\frac{W_{sensor}+F_{hd}}{W_{sensor}}\right)g$$
(6)$$a_{max}\geq -\left(\frac{W_{sensor}+F_{hd}}{W_{sensor}}\right)g$$
where *F_hd_* is the hold-down force, *W_sensor_* is the weight of rolling sensor, *g* is the gravitational acceleration, *a* is the vertical acceleration of rolling sensor, and *a_max_* is the maximum sensor acceleration to maintain sensor coupling. 

For cross-check, an accelerometer was attached onto the new sensor, and the time histories of acceleration while rolling at different speeds were measured (see Figure 3). Dynamic loads were not applied in order to check the effect of rolling speed on sensor decoupling alone (i.e., not by RDD dynamic loading). In Figure 3, with only a few exceptions along the time record, it was found that the new sensor produces negative accelerations less than −5.2 g at all tested speeds. These measurements illustrated that the rolling sensor stayed coupled with the pavement surface at all speeds. 

The magnitude of rolling noise around 30 Hz and the SNR under RDD dynamic loading were then investigated. In Figure 4, time-history raw data (5 s window) under RDD’s 30 Hz dynamic loading were collected along asphalt pavement at varied test speeds from 1.6 to 8.0 km/h (1 to 5 mph), and then converted into the frequency domain through fast Fourier transform (FFT). The authors selected the time histories with the same level of RDD deflection signal under a 30 Hz sinusoidal dynamic peak-to-peak load of ±13.3 kN (±3 kips). As test speed increases, the level of rolling noise increases and the high-frequency components significantly increase. The first and second harmonics were clearly identified at low testing speeds (see Figure 4a), but these harmonics get masked by the increased rolling noise (see Figure 4b–e). The SNRs at 30 Hz calculated from the FFT spectra were also computed and presented in the figure. As the test speed increases, rolling noise increases and the SNR decreases. 

In addition, field performance of both old and new sensors was tested on a jointed concrete pavement, which generally causes lower SNR than hot mix asphalt (HMA) because of stiffer and thicker concrete materials. During testing, no dynamic loading was applied to estimate the level of noise deflection, which is fake deflection by 30 Hz rolling noise that cannot be filtered out by the notch-pass filter used in RDD signal processing. The 30 Hz noise deflections of both old and new rolling sensors (6 and 12 in. diameters, respectively) are compared in Figure 5 with mean (μ) and standard variation (σ). It was obvious that the new sensor generated much lower noise deflection. This trend became more significant as the speed increased. Additionally, the noise comparisons at different speeds for the new sensor are compared in Table 3. Based on the authors’ experiences, noise deflections less than 2 mils do not badly affect data quality because the RDD loading is large enough to produce sufficiently high SNR in JCP sites.

## 4. Enhanced RDD Data Processing

The improved data analysis provides three major benefits over the old data analysis, being improved spatial resolution, distance-based deflection profile, and continuous SNR and noise profiles along a tested path. The flow chart of the improved data processing is shown in Figure 6. There were three additional steps added to the existing procedure, which are in the blue box in the figure, including: (1) Determination of the SNR of raw data collected over a selected distance interval (using FFT), (2) averaging the filtered data over the selected distance interval, and (3) construction of a moving-average distance-based profile.

### 4.1. Effect of the Filter Design Parameters

A sensitivity analysis was performed to evaluate the effect of the filter design parameters (BW_20_, t_90_, and N_avg_) on the filter performance and deflection profiles. BW_20_ and t_90_ are the design parameters for the IIR filter, whereas N_avg_ is for the FIR filter. As discussed earlier, when BW_20_ increases, the filter rejection-band attenuation decreases but the filter settling time increases, which adversely “smears” the RDD results. The relationships of the pole position and BW_20_ and t_90_ were checked and the results are presented in Figure 7a. Note that the performance of the FIR filter has no significant influence on t_90_ of the composite filter [15]. Figure 7b shows t_90_ of the IIR filter when different pole values are used. Bay and Stokoe [15] proposed that a reasonable t_90_ is in the range of 0.25 to 2.0 s, and the raw data restored over the t_90_ are then averaged by the decimating FIR filter. This strategy can be optimum for the slow speed-rolling sensor but may not be appropriate for higher test speeds. The speed of 8 km/h with t_90_ of 1 s will result in about the 2.3 m of spatial resolution, which is too large to assess cracks or poor joints.

The effect of the filter design parameters on the continuous deflection profile was also investigated. The authors used a raw data set of the first-generation sensor collected along a typical JCP at the speed of 1.6 km/h and the sampling rate of 256 Hz. With the same data set, the IIR and FIR filters were independently used to observe the effect of each design parameter. The sensitivity analyses were conducted using two data-processing schemes: (1) Scenario 1—fixed FIR filter (N_avg_ = 1.0 fs) and various IIR filters involving different BW_20_ values; and (2) Scenario 2—fixed IIF filter (BW_20_ = 3.9 Hz) and various FIR filters involving different N_avg_ values. In the IIR filters, the BW_20_ of 15.1, 7.7, 3.9, 1.9, and 1.3 Hz correspond to the t_90_ of 0.25, 0.5, 1, 2, and 3 s, respectively. The continuous deflection profiles constructed by the filtering scheme of Scenario 1 are shown in Figure 8a. The effect of t_90_ around the selected joint, Section A, is illustrated in Figure 8b. The joint deflections are almost constant until t_90_ = 1 s, and then begin to decrease thereafter. The decrease of average deflections on Section A can be explained by the “smearing” effect due to the increase of t_90_. In Scenario 2, various FIR filters involving different N_avg_ of 0.25, 0.5, 1, 2, and 3 fs (fs = 256 Hz in this case) were used with the fixed IIF filter, with the 3.9 Hz BW_20_ that corresponds to the t_90_ of 1 s. The continuous deflection profiles constructed by this data processing scheme of Scenario 2 are shown in Figure 9. It is clear that the increase of N_avg_ results in lower spatial resolutions in the continuous deflection profiles. In addition, the deflection level is lowered because greater portions of mid-slab areas are included with increasing N_avg._


### 4.2. New Data Processing Scheme

#### 4.2.1. Distance-Based Profile

The time-based method has a limitation that a typical time interval (1 or 2 s) has the possibility to miss critical features, such as cracks and joints, when the test speed increases. Three additional modifications were made. (1) The sampling rate was increased from 256 to 512 Hz so that a sufficient number of data points were recorded over a selected distance interval at higher test speeds. Subsequently, the IIR and FIR filters were then separately applied to raw data. The IIR filter was first applied to attenuate noise signals. BW20 in the range of 3.9 to 15.1 Hz could be selected depending on the level of rolling noise in the raw data. (2) The desired distance interval (or spatial resolution) was selected by users. If higher signal-to-noise ratios (SNRs) were obtained in raw RDD data, shorter spatial resolution with wider BW_20_ could be selected. (3) The FIR decimating filter was applied to the filtered data by the IIR filter over the selected distance interval for averaging the filtered data. In this way, the number of data points averaged over the same distance interval was different for each interval. The deflection profiles processed by the time- and distance-based reporting methods are compared in Figure 10. The data set was collected along a JCP. In the figure, 1 s t_90_ was used for the time-based profile, while the distance interval of 300 mm was used for the distance-based profile. As seen in Figure 10, slower speed causes denser data points, while higher speeds creates less spatial density. On the other hand, the deflection-based profile presented uniform distribution of deflection data along the tested path. 

#### 4.2.2. Moving Average Profile

A moving average technique is then applied to a distance-based deflection profile. In this technique, the distance interval over which raw RDD data are processed moves along the tested path with a selected distance increment, which is referred to as delta (Δ) in Figure 11a. Two adjacent distance intervals are overlapped with the length, in which the distance interval is minus the distance increment (Δ). For instance, a 0.6 m (2 ft) distance interval continues to move with a distance increment of 0.3 m (1 ft). The overlapped length between two adjacent distance intervals is 0.3 m (1 ft). As the distance interval approaches a transverse joint, the magnitude of deflection increases, and the maximum deflection takes place when the center of the distance interval is located at the joint. A graphical explanation on how to construct the moving average profile is shown in Figure 11a. 

The performance of a typical distance-based profile and the moving average profile was compared (see Figure 11b). A raw data set collected by the new sensor at 4.8 km/h was used and the distance interval of 0.5 m (1.5 ft) and its delta of 0.15 m (0.5 ft) were used for the analysis. As shown in Figure 11b, the difference of the joint deflections is about 51 μm (2 mils), which is about 18% of the “true” deflection. This difference illustrates that the RDD prototype data processing mostly underestimates deflections at joints and cracks. An increase of distance interval decreases the deflection difference at joints and mid-slabs (center area of slabs) because of the “averaging effect”. The results illustrate that the moving average technique better identifies transverse joints/cracks and estimates their deflection in a more accurate manner. In conclusion, this moving average technique can enhance the spatial resolution of deflection data but do not sacrifice the performance of signal filtering with the increasing of the test speed.

#### 4.2.3. Continuous Signal-To-Noise and Distortion Ratio (SINAD) and Noise Profiles

The SNR is a good indicator of reliability of the RDD data. Regardless of the filter performance, such as BW_20_, low SNR data cannot be effectively filtered out. In other words, assuming the loading frequency of 30 Hz, the 30 Hz noise cannot be filtered out. A continuous profile of SNR and/or noise measure along a tested path can provide the confidence level of data quality. For instance, a section involving low SNR physically means the data with poor quality due to low level of RDD signals or high level of noise. Thus, engineers should be aware of the poor data quality in interpreting the processed deflection profile. 

In this study, a signal distortion was also considered; thus, the signal-to-noise and distortion ratio (*SINAD*) was used instead of using SNR alone. Total harmonic distortion (*THD*) plus noise (*N*) is also an important indicator showing the quality of RDD signal. The *THD* is the measurement of the harmonic distortions and it is defined as the ratio of the sum of the powers of all harmonic components to the power of the fundamental frequency. When the input is a pure sine wave, the measurement is most commonly the ratio of the sum of the powers of all higher harmonic frequencies to the power at the first harmonic, or fundamental, frequency:(7)$$THD=\left(\frac{P_{2}+P_{3}+P_{4}+\cdot \cdot \cdot +P_{\infty }}{P_{1}}\right)=\frac{P_{total}-P_{1}}{P_{1}}$$
where *P*_n_ is the average power of nth harmonic (n = 1 is the fundamental frequency (30 Hz). *THD + N* represents the level of noise and signal distortion compared to the signal, and is defined as:(8)$$THD+N=\frac{\left(P_{total}-P_{1}\right)+P_{noise}}{P_{1}}$$

The *SINAD* is a measure of the quality of a signal, defined as: (9)$$SINAD=\frac{P_{signal}+P_{noise}+P_{distortion}}{P_{noise}+P_{distortion}}$$
where *P* is the average power of the signal, noise, and distortion components. The *SINAD* is usually expressed in a decibel (dB) scale while the *THD* and *THD + N* are expressed in percentage as distortion factors. It is important to note that the *SINAD* reading cannot be less than 1.0. 

In the RDD data analysis procedure, the noise concentrated near the RDD operating frequency (around 30 Hz) is a major concern, because the IIR filter (notch-pass filter) cannot remove noise the same as the operating frequency. Rolling noise and signal distortion presented outside this range will be effectively filtered out. Thus, the *SINAD, THD*, and *THD + N* determined over the range of 20 to 40 Hz were used in this study, defined as: (10)$$SINAD_{20–40\;\mathrm{Hz}}\left(dB\right)=10\cdot \log_{10}\left(\frac{P_{signal,30\;\mathrm{Hz}}+P_{noise,20–40\;\mathrm{Hz}}+P_{s=distortion,20–40\;\mathrm{Hz}}}{P_{noise,20–40\;\mathrm{Hz}}+P_{s=distortion,20–40\;\mathrm{Hz}}}\right)$$
(11)$$THD_{20–40\;\mathrm{Hz}}=\frac{P_{2}+P_{3}+P_{4}+\cdots +P_{\infty }}{P_{1}}=\frac{P_{total}-P_{1}}{P_{1}}\approx 0$$
(12)$$THD+N_{20–40\;\mathrm{Hz}}\left(\%\right)=THD_{20–40\;\mathrm{Hz}}+\frac{P_{noise,20–40\;\mathrm{Hz}}}{P_{1}}\approx \frac{P_{total,20–40\;\mathrm{Hz}}}{P_{1}}$$
where *P_1_* is the power of the first harmonic (fundamental frequency of 30 Hz), and *P_noise,20–40Hz_* is the average power of rolling noise in the frequency range of 20 to 40 Hz. 

Continuous *SINAD, THD,* and *THD + D* profiles were constructed with RDD data sets of which the first-generation rolling sensor was tested at 1.6 km/h (0.45 m/s) along a JCP at the Texas Department of Transportation (TxDOT) Flight Service Facility (FSF). Figure 12a shows the continuous RDD deflection profile measured along JCP. Continuous *SINAD* and *THD + N* profiles are presented in Figure 12b,c, respectively. In these figures, the thicker slabs create lower levels of pavement deflections, resulting in lower *SINAD* but higher *THD + N*. The field visual survey also confirmed that the 40 cm (16 in.) thick slabs involve concrete surface tining and a wider joint opening that causes higher rolling noise. The *THD* is almost zero over the entire path because the harmonic distortions outside the frequency range of 20 to 40 Hz were not considered in this analysis. In addition, several locations in the section of 40 cm thick slabs involve the *THD + D* measurements exceeding 100%, which means that the average power of noise plus distortion over that frequency range (from 20 to 40 Hz) is greater than the power of RDD’s 30 Hz signal.

In addition, the performances of old and new generation sensors are compared. The continuous *SINAD* and *THD + N* profiles are presented in Figure 13a,b, respectively. The *SINAD* profiles presented in Figure 13a show that the new sensor made significant improvement of *SINAD*. The average *SINAD* values of the old and new sensors are 8.3 and 13.0 dB, respectively. In addition, the *THD + N* profiles in Figure 13b showed that the noise was apparently reduced at the same speed of 1.6 km/h. 

## 5. Conclusions

RDD’s continuous deflection profile is a powerful tool in evaluating the existing structural condition of pavements, which is a critical input for pavement forensic investigation and rehabilitation studies. In the previous study [2], a mechanical design (with theoretical perspective) and simple field performance check was conducted to explore the feasibility of a speed-improved sensor design. This study provides an enhanced digital filter and data processing scheme for the speed-improved sensor so that the RDD system is enhanced in both hardware and software; thus is able to speed up to 8.0 km/h. In addition, some missing research components from the previous study [20] are presented herein; for example, RDD’s signal SNR under operation, noise deflection at varied speeds, sensor decoupling (hold-down force vs. allowable acceleration). The new sensor design employs a larger diameter, wider wheel, and a hold-down force system. The new data processing utilizes optimized digital filter design parameters, a distance-based reporting method (with moving average), and it overcomes limitations of the time-based method that includes lower spatial resolution at increased test speeds, variation of deflection-reporting interval due to the speed variation, and the potential of missing key features (i.e., cracks and joints). The conclusions drawn from this study are summarized as below. First, the benefits of the new sensor over the previous sensors include a lowered level of rolling noise, improved sensor coupling at higher test speeds, and higher SNR. The sensor wheel was designed to improve SNR by maximizing the signal (induced pavement deflection) and attenuating rolling noise. In JCP sites, the new sensor (two-wheel system) would be powerful because accurate positioning of sensor and point-like deflection are prerequisites for the assessment of load transfer conditions at joints and cracks. Second, the distance-based profile is more beneficial compared to the time-based method, particularly for the JCP assessment. The moving average technique can minimize the underestimation of deflection, which can be up to 20% underestimation in deflection at joints. In addition, the speed variation does not affect the spatial resolution in the deflection profile. Third, the *SINAD* and noise profiles can be used for evaluating the reliability of raw data at higher test speeds. If rolling noise is greater than the RDD signal, the notch-pass filter would not be effective in removing noise signals. Identifying the areas of low-quality data can remove the possibility of false alarms with high deflection peaks, and also can help engineers better interpret the analyzed data. 

## Figures and Tables

**Figure 1 materials-12-01653-f001:**
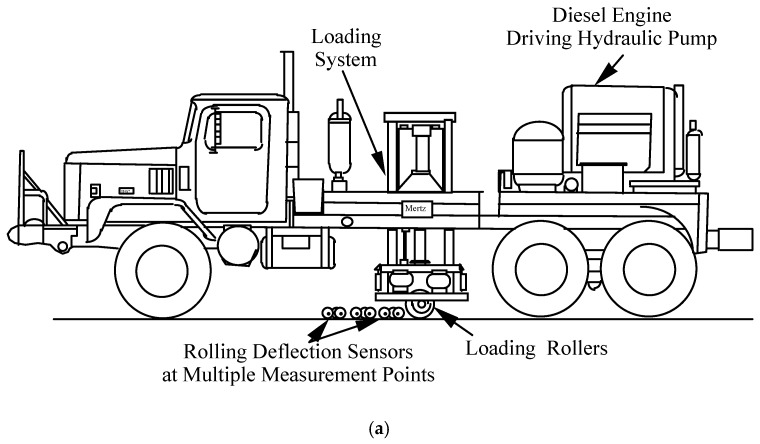
General description of the RDD: (**a**) A sideview of the RDD, (**b**) a typical RDD rolling sensor array (after Bay and Stokoe [15]).

**Figure 2 materials-12-01653-f002:**
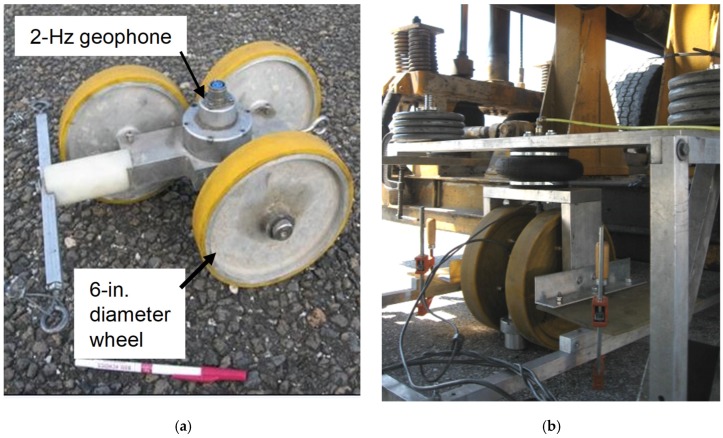
The rolling sensors of the RDD: (**a**) Old rolling sensor, and (**b**) new rolling sensor.

**Figure 3 materials-12-01653-f003:**
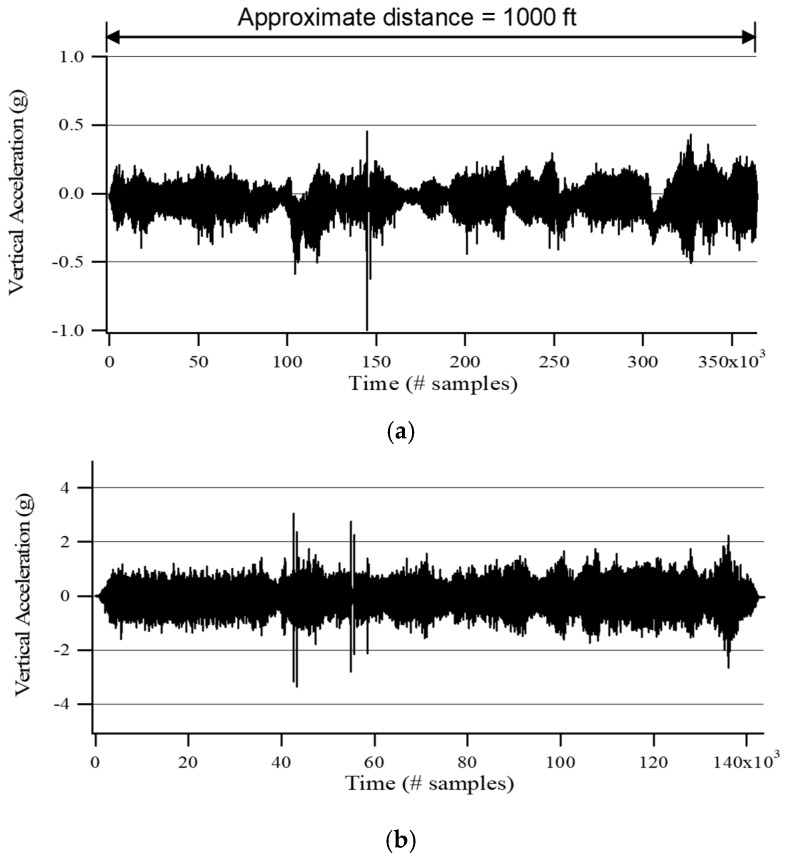
Vertical acceleration of the new generation sensor (Sensor #1) at different test speeds collected along asphalt pavement with no RDD dynamic load. (**a**) 1.6 km/h (1 mph); (**b**) 4.8 km/h (3 mph); (**c**) 8.0 km/h (5 mph).

**Figure 4 materials-12-01653-f004:**
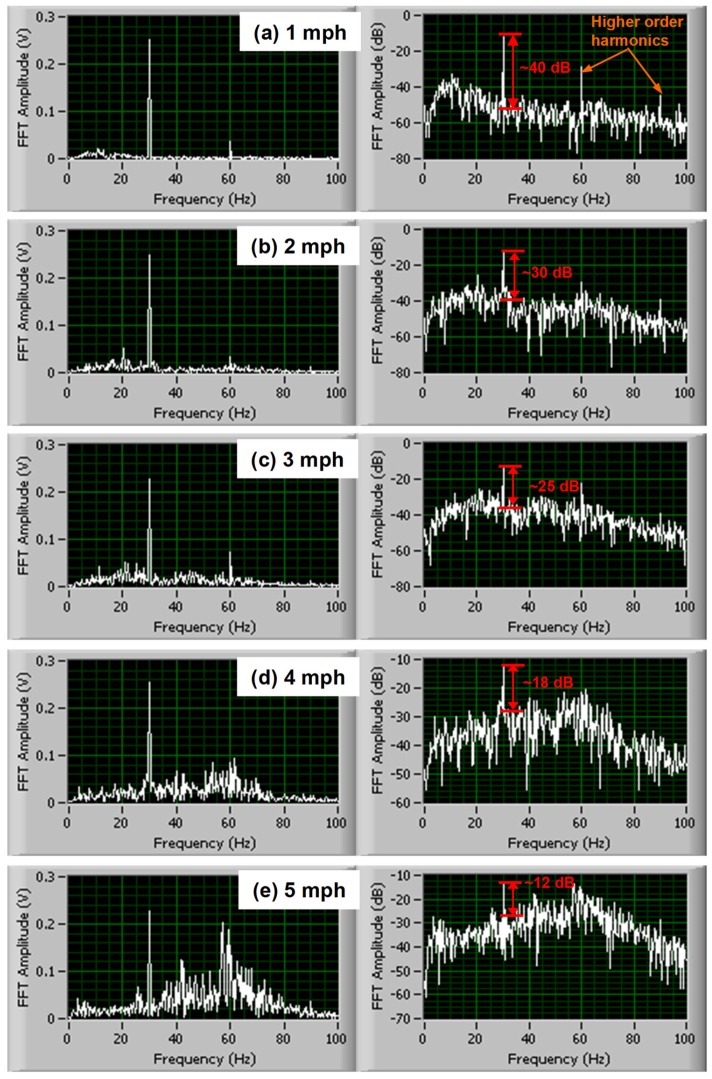
Signal-to-noise ratio (SNR) and rolling noise with varied test speeds (5 s window) (note: new rolling sensor (Sensor #1), 5 psi sensor hold-down pressure, RDD operating frequency of 30 Hz, and 3 kip peak dynamic loading). (**a**) 1 mph; (**b**) 2 mph; (**c**) 3 mph; (**d**) 4 mph; (**e**) 5 mph.

**Figure 5 materials-12-01653-f005:**
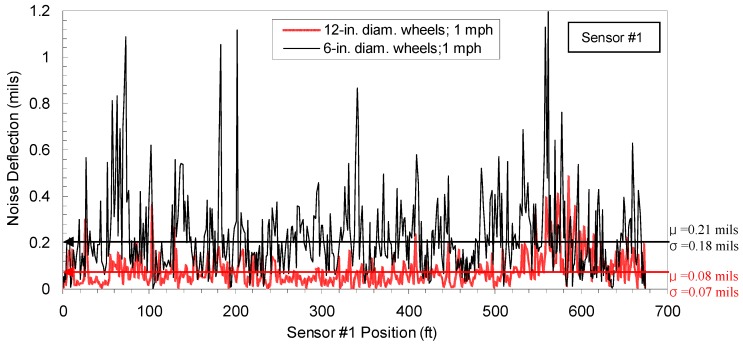
Noise-deflection profiles of the old and new rolling sensors, collected along a jointed concrete pavement at the speed of 1 mph (note: the wheel diameters of the old and new sensors are 6 and 12 in. diameters, respectively).

**Figure 6 materials-12-01653-f006:**
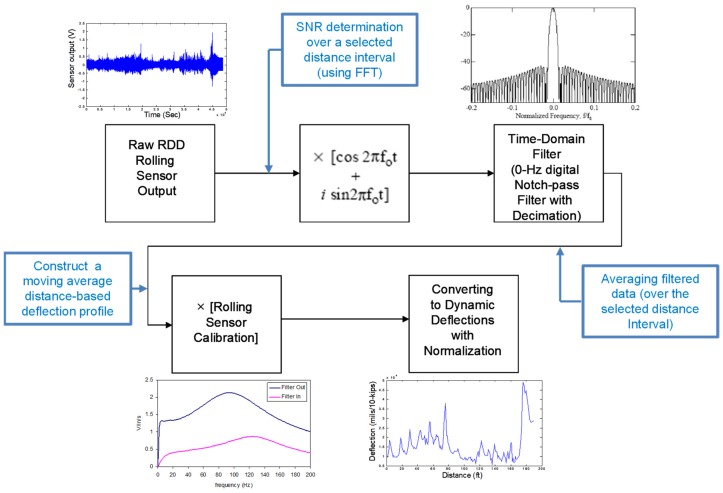
Procedure of new data processing for the speed-improved rolling sensors (note: three additional steps, shown as blue boxes, were added for the new data processing).

**Figure 7 materials-12-01653-f007:**
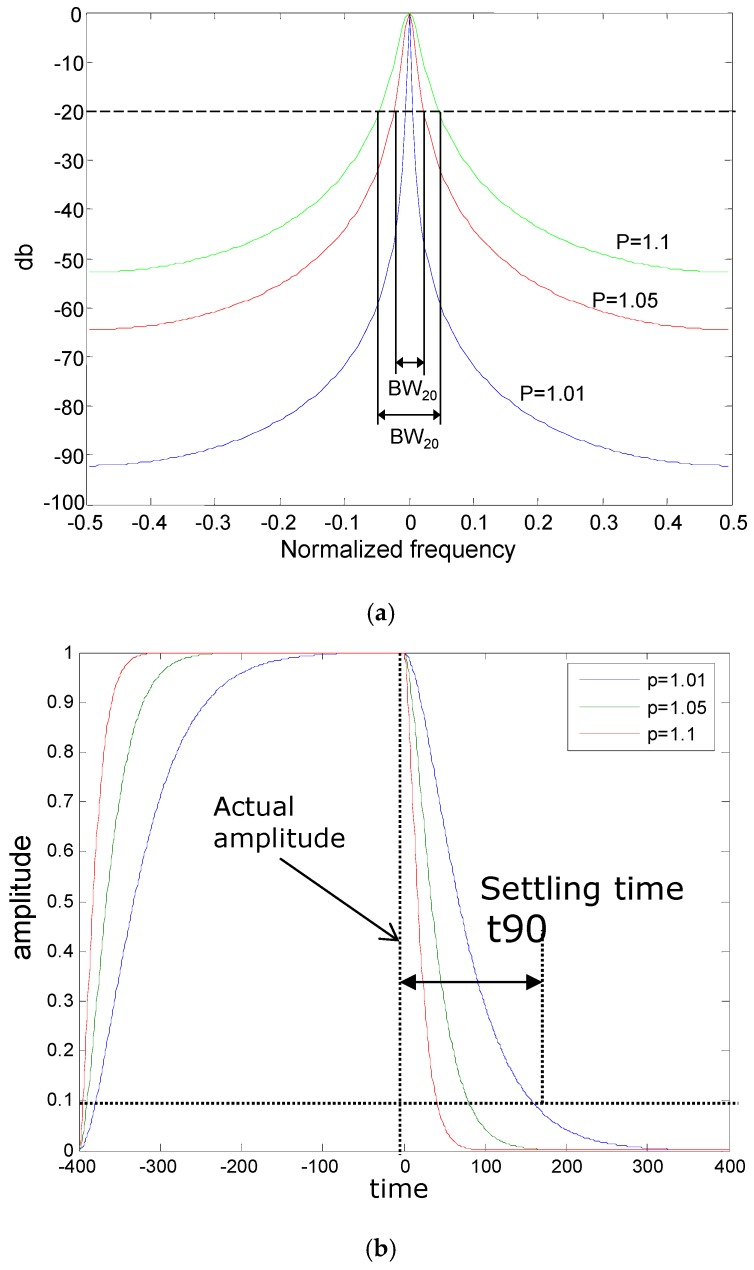
Performance of the IIR digital filter: Filter responses of the 0 Hz IIR digital filters with varied poles in (**a**) frequency and (**b**) time domains.

**Figure 8 materials-12-01653-f008:**
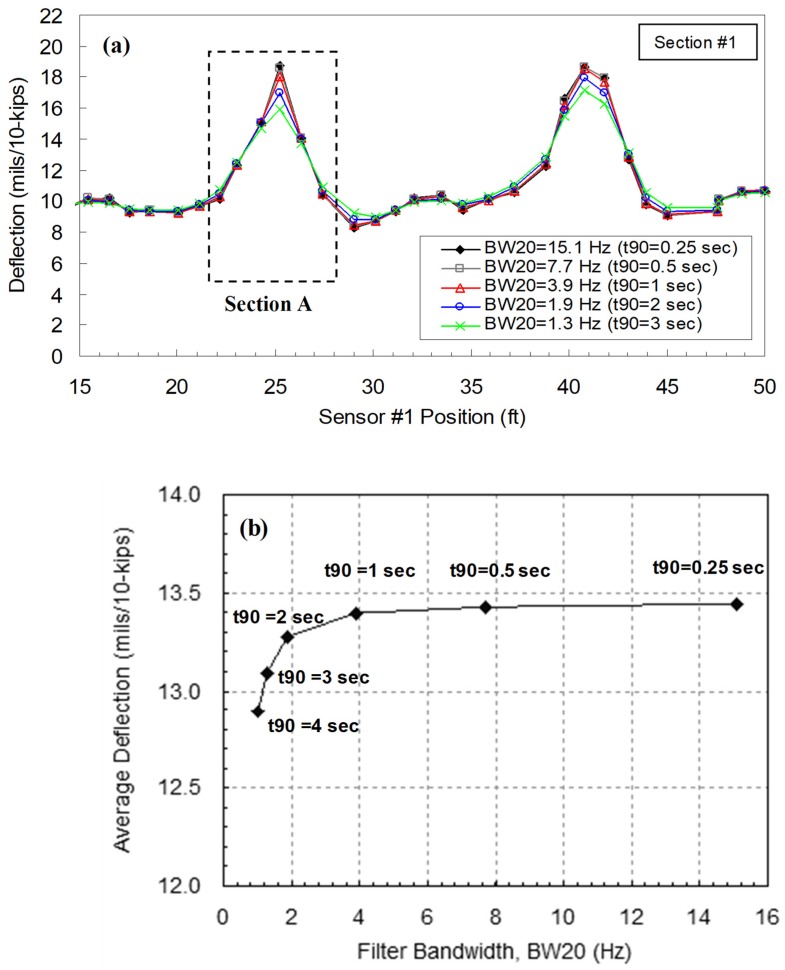
RDD deflection measurement: (**a**) Deflection profiles constructed by various IIR filters with different BW_20_ (but fixed FIR filter); and (**b**) relationship between IIR filter BW_20_ and average deflection over Section A (showing the effect of filter settling time, t_90_).

**Figure 9 materials-12-01653-f009:**
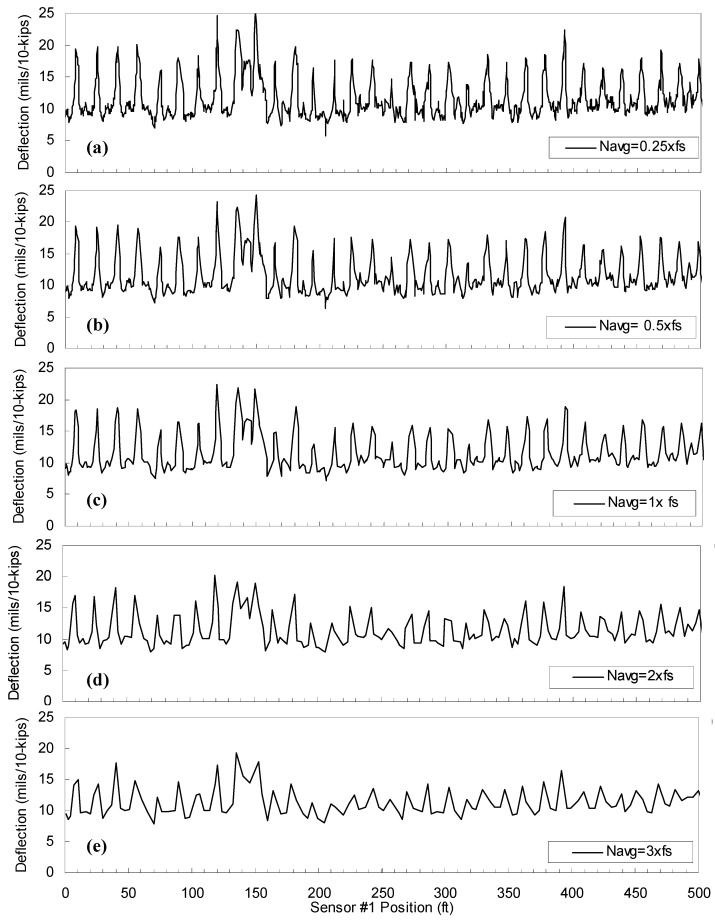
Continuous deflection profiles (Sensor #1) constructed by various FIR filters with different N_avg_ but the same IIR filter (BW_20_ = 3.9 Hz). (**a**) 0.25× f_s_; (**b**) 1× f_s_; (**c**) 2× f_s_; (**d**) 3× f_s_.

**Figure 10 materials-12-01653-f010:**
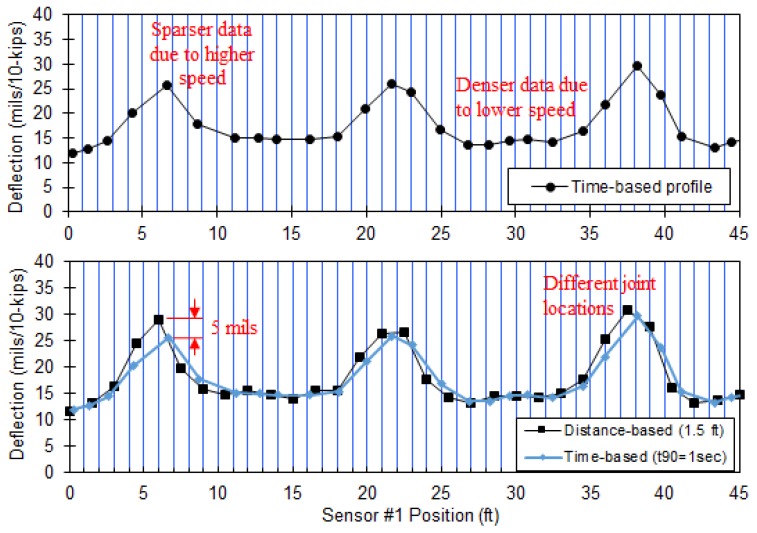
Continuous deflection profiles constructed by the time- and distance-based methods (note: data set collected along a JCP at 1.6 km/h (1 mph)).

**Figure 11 materials-12-01653-f011:**
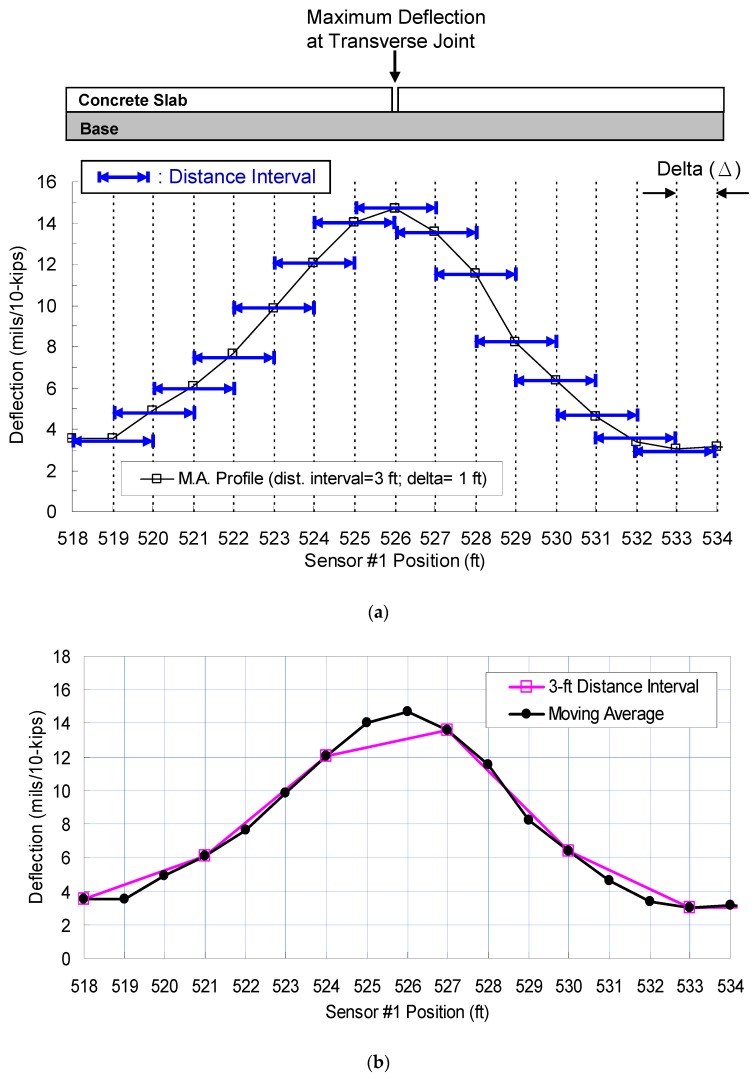
Continuous moving-average profile around a transverse joint: (**a**) Procedure to construct the moving average profile, and (**b**) the comparison of the moving average and the distance-based profile (with 1.5 ft spatial interval).

**Figure 12 materials-12-01653-f012:**
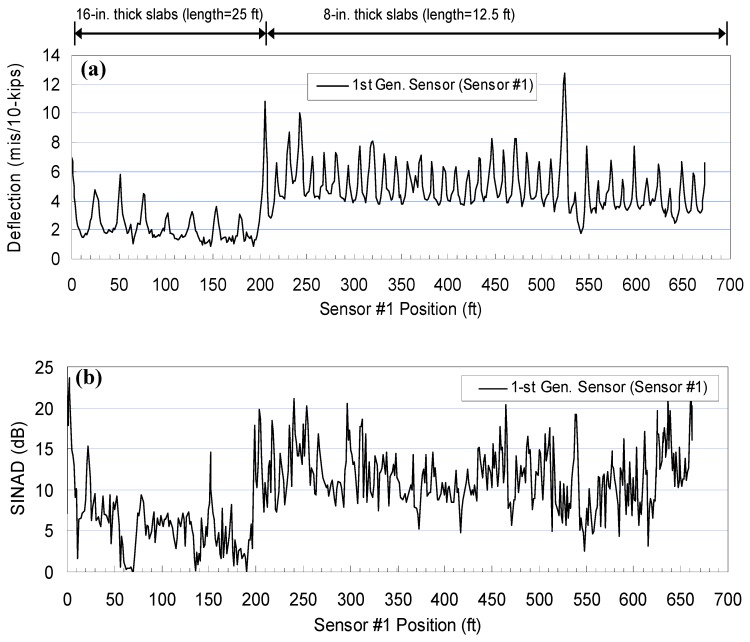
RDD measurements using the old rolling sensor (referred as first generation sensor) at 1 mph along the JCP: (**a**) Continuous deflection profile, (**b**) continuous *SINAD* profile, and (**c**) continuous *THD + N* and *THD* profiles.

**Figure 13 materials-12-01653-f013:**
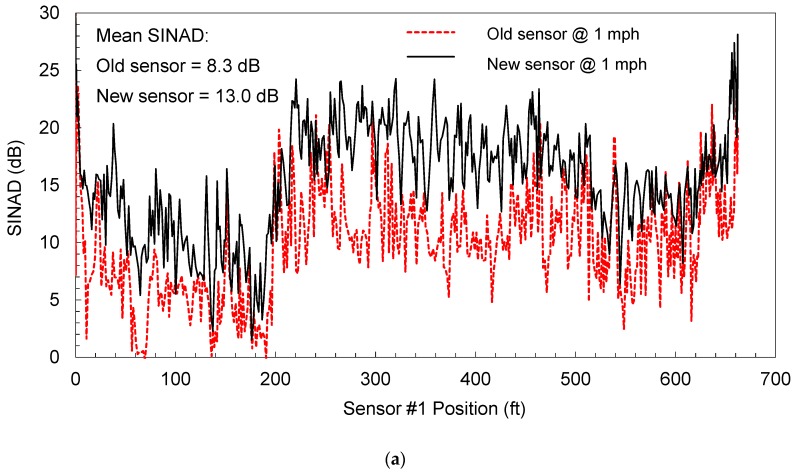
Comparison of noise characteristics of both old and new rolling sensors: (**a**) *SINAD* profile and (**b**) *THD + N* profile along a JCP site.

**Table 1 materials-12-01653-t001:** Testing devices for continuous pavement deflection profiling.

Testing Methods	Loading/Sensing Types	Testing Speed (mph)	Sensor Accuracy (mils)	Applied Loading (kips)	Data Spatial Resolution (ft)	Manufacturer	Updated References
Traffic Speed Deflectograph (TSD)	Moving load/ non-contact (laser doppler sensor)	45	±4 mils/s	11	33	Greenwood Engineering	[1,2,3,4,5,6]
Rolling Wheel Deflectometer (RWD)	Moving load/ non-contact (laser displacement sensor)	55	±2.75 mils	18 (fixed)	100 to 500	Applied Research Associates	[6,7,8,9,10]
Airfield Rolling Weight Deflectometer (ARWD)	Moving load/ non-contact (laser displacement sensor)	20	0.8 mils	9	90	Dynatest Consulting and Quest Integrated	[6,11,12]
Rolling Deflection Tester (RDT)	Moving load/ non-contact (laser displacement sensor)	36	±10 mils	8 to 14	66	Swedish National Road Administration and VTI *	[13,14]
Rolling Dynamic Deflectometer (RDD)	Sinusoidal load/ Contact (geophone)	1 to 3	0.05 mils	10 kips static ±5 kips dynamic	1.5 to 3	The University of Texas at Austin	[15,17,18,19,20,21,22,23,24,25,26]

(* VTI = The Swedish National Road and Transport Research Institute)

**Table 2 materials-12-01653-t002:** Statistics of noise-level deflection measurements.

Speed	Mean Noise Deflection, μ	Standard Deviation, σ
1 mph	0.07 mils (2 μm)	0.08 mils (2 μm)
2 mph	0.29 mils (7 μm)	0.21 mils (5 μm)
3 mph	0.50 mils (13 μm)	0.43 mils (11 μm)
4 mph	0.75 mils (19 μm)	0.84 mils (21 μm)
5 mph	1.09 mils (28 μm)	1.19 mils (30 μm)

**Table 3 materials-12-01653-t003:** Theoretical maximum acceleration with different levels of hold-down. Pressure in air spring.

Additional Hold-Down Pressure (psi)	Additional Hold-Down Force (lbs)	Maximum Allowable Acceleration (g)
0	0	−1
1 (6.9 kPa)	19 (8.6 kg)	−1.8
2 (13.8 kPa)	38 (17.2 kg)	−2.7
3 (20.7 kPa)	57 (25.8 kg)	−3.5
4 (27.6 kPa)	76 (34.5 kg)	−4.4
5 (34.5 kPa)	95 (43.1 kg)	−5.2

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
