# Peer review of "Enhanced Sensing and Data Processing System for Continuous Profiling of Pavement Deflection"

_materials, 2019, doi:10.3390/ma12101653_

Round 1

Reviewer 1 Report

I am sorry to send you a review that could appear contradictory.

The main point is: your literature analysis reported in the "Introduction" is extremely outdated. You are missing the last 15 years improvements in the field of pavement deflections acquisition.

If you analyze the paper referenced in the literature there is NO paper published in the last 5 years. The equipment background is based on papers published before 2000 (references 1 to 6).

The analysis of the rolling weight deflectometers available in the market is missing the Greenwood Traffic Speed Deflectometer and the Dynatest Raptor: there is a huge literature on the subject of continuous deflection acquisition you are completely missing in your paper.

Just have a look at the Transportation Research Board website and check in the papers presented at the last 5 Annual Meetings: you will find so many papers ignored in your study.

I do not understand the importance of improving the acquisition speed from 1.6km/h to 8km/h when there are available contactless equipment able to perform 10 times faster.

What I suggest is:

 - to update the current status of equipment available on the market (Table 1)

 - to present the accuracy of the system you improved compared to the new systems

 - to present the benefits of using sensors in contact with the pavement

It seems to me that your study arrives 15 years late in the technical scenario of pavement deflection data acquisition. In my view there is the absolute need to solve this problem in order to avoid your efforts appear as a useless exercise.

Some editing suggestions:

Raw 34: please remove the double comma after (RWD),,

Raw 87: Figure 1: the (c) figure seems not to be present

Raw 113: Equation 1: please correct the formula

Raw 117: this sentence should be moved after equation (2)

Raw 135: please remove the dot "below. (1)"

Raw 269: please correct the  Figure 8(b). It is not possible to understand what the different arrows are indicating

Raw 379: Figure 13(b), please correct what is reported in the legend "rough surface &" 

Author Response

Reviewer #1

The main point is: your literature analysis reported in the "Introduction" is extremely outdated. You are missing the last 15 years improvements in the field of pavement deflections acquisition.

If you analyze the paper referenced in the literature there is NO paper published in the last 5 years. The equipment background is based on papers published before 2000 (references 1 to 6).

The analysis of the rolling weight deflectometers available in the market is missing the Greenwood Traffic Speed Deflectometer and the Dynatest Raptor: there is a huge literature on the subject of continuous deflection acquisition you are completely missing in your paper.

Just have a look at the Transportation Research Board website and check in the papers presented at the last 5 Annual Meetings: you will find so many papers ignored in your study.

It seems to me that your study arrives 15 years late in the technical scenario of pavement deflection data acquisition. In my view there is the absolute need to solve this problem in order to avoid your efforts appear as a useless exercise.

Response: Thanks for the valuable comments. The literature analysis has been significantly updated and enhanced based on the comments.  

I do not understand the importance of improving the acquisition speed from 1.6km/h to 8km/h when there are available contactless equipment able to perform 10 times faster.

Response: More clear explanations are provided below. Please note that the RDD is mainly used for project-level studies. The RDD employs contact-type loading and sensing systems so that the accuracy and data spatial resolution are better than those contactless technologies (with non-contacting sensors). Those contactless equipment employs moving loading system (not sinusoidal loading), thus no use of digital band-pass filter that can remove unwanted noise signals. The data spatial resolution is not sufficient for project level studies, thus they are used only for network-level studies not project-level studies. The RDD has been extensively used in TxDOT’s project-level studies (e.g. pavement forensic investigation, identification of poor areas (or slabs) to be repaired, joint load transfer assessment, selection of optimum rehabilitation strategy, etc.) [17-20; 22-23]. See the updated reference.

What I suggest is:

 - to update the current status of equipment available on the market (Table 1)

Response: Yes, it was updated and most recent references are added.

 - to present the accuracy of the system you improved compared to the new systems

 - to present the benefits of using sensors in contact with the pavement

Response: Thanks for the comments. Table 1 was updated and a description was added.

Please note that the RDD is mainly for project-level studies not network-level studies. Although the RDD is a continuous pavement testing device, its efficiency (e.g. cost, time, etc.) and accuracy are compared with FWD. Other continuous deflection testing devices (e.g. TSD, Raptor) are used for network-level projects and not suitable for project-level studies due to insufficient spatial resolution of deflection data.  In addition, the RDD employs the contact loading and sensing systems in order to gain more accurate measurement (induced pavement response) and higher spatial resolution of deflection data. As seen in the revised table (Table 1), the contact sensors are beneficial in two areas: sensor accuracy (laser sensor vs. geophone) and the spatial resolution. Lastly, the most powerful benefit of the RDD is the use of digital filter that can filter out unwanted noise signals. This is possible due to sinusoidal loading (vibration with a single frequency) through the contact-type loading. This strength is only applicable to the RDD not to other moving loading system (e.g. TSD, Raptor, etc.).   Introduction was significantly revised including more clear descriptions.  

Some editing suggestions:

Raw 34: please remove the double comma after (RWD),,

Response: Yes, corrected.

Raw 87: Figure 1: the (c) figure seems not to be present

Response: It is corrected. Figure 1(c) is not presented and the subfigure title is removed. Instead, a description about the forcing function is added.  

Raw 113: Equation 1: please correct the formula

Response: Yes, corrected.

Raw 117: this sentence should be moved after equation (2)

Response: Yes, corrected. Z is defined in Line 114 so removed.

Raw 135: please remove the dot "below. (1)"

Response: Yes, corrected.

Raw 269: please correct the  Figure 8(b). It is not possible to understand what the different arrows are indicating

Response: The figure was corrected. Sorry, it was an error while changing the format to MDPI Journal.

Raw 379: Figure 13(b), please correct what is reported in the legend "rough surface &"

Response: Corrected.

Reviewer 2 Report

In the article, the figures 3, 4, 6, 8 is missing for review. The article is long but interesting.

Author Response

In the article, the figures 3, 4, 6, 8 is missing for review. The article is long but interesting.

Response: Sorry for the error. Please see the message from the journal.

“The comments that some figures are missing in text, I am sorry it was caused by us while converting it into PDF. I have uploaded the correct version and will explain it to the reviewer, you can clarify it in your response to reviewer too.”

Reviewer 3 Report

The figures 2, 3, 4, 6 and 8 do not have any readable format in my pdf version of the article.

I believe that the article is worthy and important for the readers. My doubt is: what was improved from the work described in your reference [20]. Actually the objective then (2014 first published, 2012 first submitted) seems pretty much the same then now. Can you explain and consider this explanation in your conclusions now?

Author Response

Reviewer #3

The figures 2, 3, 4, 6 and 8 do not have any readable format in my pdf version of the article.

Response: Sorry for the error. Please see the message from the journal.

“The comments that some figures are missing in text, I am sorry it was caused by us while converting it into PDF. I have uploaded the correct version and will explain it to the reviewer, you can clarify it in your response to reviewer too.”

I believe that the article is worthy and important for the readers. My doubt is: what was improved from the work described in your reference [20]. Actually the objective then (2014 first published, 2012 first submitted) seems pretty much the same then now. Can you explain and consider this explanation in your conclusions now?

Response: Thank you for the valuable comment. The conclusion was revised. Here is my short explanation.  The previous paper [2] is about the hardware development in sensor, particularly mechanical design (with theoretical perspective) and physical check in the field to see the feasibility of the new sensor in both asphalt and concrete pavements. With the new sensor, the old data processing was used. However, this paper provided newly developed software, which is digital filter and data processing scheme, to improve the accuracy and performance of those new sensors. In addition, some missing research components from the previous study [20] are presented herein. For example, we further investigated RDD’s signals under operation (see Fig. 4), noise deflection at varied speeds, sensor decoupling (hold-down force vs. allowable acceleration), and so on. This enhanced data processing is very necessary for characterizing cracks/joints. At the end, both newly developed hardware and software should be used for actual projects.

Also the title is changed to “Enhanced sensing and data processing system for continuous profiling of pavement deflection”

Round 2

Reviewer 1 Report

The reviewed version is adequately facing the Rolling Weight Deflectometer status and the research activity undertaken seems to be better introduced than in the previous version of the paper.